# Hybridization Effects on Bending and Interlaminar Shear Strength of Composite Laminates

**DOI:** 10.3390/ma15041302

**Published:** 2022-02-10

**Authors:** Alice Monjon, Paulo Santos, Sara Valvez, Paulo N. B. Reis

**Affiliations:** 1Department of Aeronautic Engineering, University of Beira Interior, 6201-100 Covilhã, Portugal; alicemonjon@hotmail.com; 2Department of Electromechanical Engineering, C-MAST, University of Beira Interior, 6201-100 Covilhã, Portugal; paulo.sergio.santos@ubi.pt (P.S.); sara.valvez@ubi.pt (S.V.); 3Department of Mechanical Engineering, CEMMPRE, University of Coimbra, 3030-788 Coimbra, Portugal

**Keywords:** Polymer–Matrix Composites (PMCs), hybridization, mechanical properties, mechanical testing

## Abstract

Fiber-reinforced composites are gradually replacing the traditional materials in many engineering applications. However, for many applications these materials are still unsuitable, due to their lack of toughness. In this context, hybridization is a promising strategy in which two or more types of fiber are combined to obtain a better balance of mechanical properties compared to non-hybrid composites. Therefore, the main goal of this work is to study the hybridization effect on the static performance and interlaminar shear strength. For this purpose, carbon, glass, and Kevlar fibers were used and combined in different proportions. It was possible to conclude that there is an ideal value of fiber content to maximize both properties and, depending on the type of fiber, they should be placed specifically on the compression or tensile side. For example, for composites involving carbon and glass fibers the latter must be placed on the compression side, and for a value of 17% by weight the flexural strength decreases by only 2.8% and the bending modulus by around 19.8%. On the other hand, when Kevlar fibers are combined with glass or carbon fibers, the Kevlar ones must always be placed on the tensile side and with an ideal value of 13% by weight.

## 1. Introduction

Fiber-reinforced composites are one of the most remarkable families of materials for technological and structural applications. Nowadays, they are widely used in sectors such as the automotive and military industries, the renewable energy industry, infrastructures, medicine, and sports, but their sector of predilection is the aeronautical field [1,2]. In fact, these materials have the ability to be tailored for use, and a wide variety of fiber and matrix combinations are possible, opening up possibilities for many applications.

Hybrid composites, for example, are materials that combine two or more types of fibers in a same resin matrix. With this strategy it is possible to obtain more balanced materials in terms of mechanical properties and, consequently, adapt them to the design requirements [3,4]. Basically, the advantages of the fibers are valued, while the weaknesses of each one of them are minimized [5,6]. For example, high-modulus fibers (such as carbon fibers) or low-elongation (LE) fibers have the advantage of providing stiffness and load carrying capacity but less elongation and compressive strength, while lower-modulus fibers (such as glass and Kevlar fibers) or high-elongation (HE) fibers are characterized by lower stiffness, higher elongation, and damage tolerance. Therefore, their combination allows one to improve the toughness, although the final strength and stiffness are inferior to those of the high modulus [7,8]. According to Swolfs et al. [7], LE and HE fibers can be combined in different modes, such as interlayer or layer-by-layer configuration (layers of two different fibers are stacked onto each other), intralayer or yarn-by-yarn configuration (two different fibers are mixed within the layers), and intrayarn or fiber-by-fiber configuration (two different fibers are mixed/commingled on the fiber level). Although it is possible to obtain more complex configurations by combining two of the three configurations, the first is the simplest and cheapest.

Regarding the bending properties, Guo et al. [9] observed for hybrid rods that a good uniformization of the carbon fiber dispersion into the glass fiber could lead to an increase in bending strength by up to 60.3% while in the bending modulus by up to 39.6%. On the other hand, these properties are also strongly dependent on the stacking sequence. Giancaspro et al. [10], for example, noted that fiberglass composites fail more easily when placed on the tensile side, while carbon fiber composites are more sensitive when placed on the compression side. Wonderly et al. [11] also reported that the ratio of compressive strength over tensile strength is different for carbon and glass fiber composites, with values around 0.34 and 0.73, respectively. In fact, Santos et al. [12] and Ghafaar et al. [13] reported that full-carbon composites have the maximum bending stress and stiffness, while full-glass fiber laminates have the lowest value, reaching a difference in bending strength of about four times [14]. Therefore, by adding carbon fibers on the tensile side of glass fiber composites, the flexural strength will increase, while on the compression side the strength will decrease [14,15]. According to Dong et al. [16], the highest flexural strength is achieved for a relative content of glass fibers in the order of 12.5% and all placed on the compressive side. In another study, the same authors observed that the flexural strength for hybrid carbon/glass composites is 40% and 9% higher than those for all-carbon and all-glass composites, respectively. Kevlar fibers, on the other hand, show a significant difference between tensile and compression strength, conditioning the flexural behavior of composites reinforced with these fibers. For example, the strain at the compressive side is larger than that at the tensile side, causing the shift of the neutral axis to the tensile side with the growing compressive yield region. When they are subjected to axial compression or bending, they may exhibit a nonlinear plastic deformation as a consequence of structural defects developed in the chain of the fibers [17]. Finally, fibers’ fracture is usually preceded by longitudinal fragmentation and splintering, a non-catastrophic failure mode that gives to Kevlar fiber-reinforced composites a high tolerance to damage from impact or other dynamic loads that is not observed for reinforced composites with glass or carbon fibers [3].

The mechanical performance of polymer composites strongly depends on their fiber/matrix interface [18,19], because a low interlaminar shear strength, for example, is usually due to poor bonding between fiber and matrix [20]. On the other hand, the presence of voids or moisture in composites also affects the interlaminar shear strength (ILSS). For example, in carbon/epoxy composites, a void content around 10% by volume reduces the ILSS by around 25% [21]. In the same context, the fiber type and its orientation have a strong influence on the interlaminar shear strength [13,20]. According to Madhavi et al. [14], the ILSS values for carbon-reinforced composites are about five times higher than those observed for glass fiber-reinforced composites. Therefore, hybridization can also be used to improve the interlaminar shear strength. Studies developed by Turla et al. [22], involving laminates reinforced with carbon fibers, glass fibers, and carbon/glass hybrids, revealed that the ILSS of the hybrid composite is significantly higher than that of glass or carbon composites. Padmanabhan et al. [23] also observed an improvement in the ILSS values with increasing thickness, due to the higher bending stiffness obtained. At the level of Kevlar fibers, they are known to present weak interfaces with different matrices, and, in terms of thermosetting matrices, epoxy resins generally lead to higher ILSS values. However, manufacturing defects are the main mechanisms that affect the interlaminar shear strength [3,22].

Nevertheless, according to Swolfs et al. [7], hybrid effects under more complex loading conditions, such as in bending, impact, and fatigue tests, are not well understood and sometimes even promote apparent contradictions. These authors even suggest further work to obtain more robust conclusions. Therefore, this work intends to consolidate the conclusions presented in the literature in terms of static bending strength and interlaminar shear strength. For this purpose, composites involving different fibers (such as carbon, glass, and Kevlar fibers) and different values of weight content were used.

## 2. Materials and Methods

Carbon fiber woven bidirectional fabric (taffeta with 195 g/m^2^), glass fiber woven bidirectional fabric (taffeta with 195 g/m^2^), and kevlar fiber woven bidirectional fabric (taffeta with 170 g/m^2^), all in the same direction, with an Ebalta AH 150 resin and IP 430 hardener were used to prepare different composite laminates. Three groups of samples were prepared by hand layup, with the stacking sequence shown in Table 1. The numbers represent the quantity of layers, while the letters represent the carbon fibers (C), Kevlar fibers (K), and glass fibers (G).

Each system was placed inside a vacuum bag and compressed in a hydraulic press with a load of 2.5 kN applied for 24 h to maintain a uniform thickness and a constant fiber volume fraction. During the first 2 h, the bag remained attached to a vacuum pump to eliminate any air bubbles introduced during the manufacturing process. Finally, following the manufacturer’s datasheet recommendations, the plates were subjected to a post-cure in an oven at 80 °C for 5 h.

The methodology described above was used to produce composite laminates with dimensions of 330 × 330 × t mm^3^, from which were obtained specimens with dimensions of 100 × 10 × t mm^3^ for the static bending tests and for the ILSS tests according to ASTM D 2344/D 2344M standard. Regarding the three-point bending (3PB) tests, they were carried out using a Shimadzu universal machine, model Autograph AG-X, equipped with a 10 kN load cell, and at least five samples for each condition were carried out according to the recommendations of the ASTM D790-03. The bending properties were obtained using the following equations:(1)$$\sigma_{f}=\frac{3\mathrm{PL}}{2{\mathrm{bh}}^{2}}$$
(2)$$E_{f}=\frac{∆{\mathrm{PL}}^{3}}{48∆\mathrm{uI}}$$
(3)$$\varepsilon_{f}=\frac{6\mathrm{Sh}}{L^{2}}$$
where P is the load, L the span length, b the width, h the thickness of the specimen, S the deflexion, I the moment of inertia of the cross section, ΔP the load range, and Δu the bending displacement range in the mid span for an interval in the linear load-displacement region of the graph. The bending modulus was obtained by linear regression of the load-displacement curves, considering the interval in the linear segment with a correlation factor higher than 95% [24,25].

In terms of ILSS tests, they were carried out according to the recommendations of the ASTM D 2344/D 2344M and the values obtained using the following equation:(4)$$\mathrm{ILSS}=0.75\;\frac{P}{\mathrm{bh}}$$
where P is the load, b the width, and h the thickness of the specimen.

The failure modes resulting from the bending tests performed for the different configurations were analyzed using a Nikon optical microscope (model SMZ-2T).

Finally, the fiber content for the different composites was also obtained by the chemical matrix digestion technique, according to ASTM D3171-15 standard, and using concentrated nitric acid to dissolve the epoxy resin. However, because some Kevlar filaments are partially dissolved in this process, the dissolution technique suggested by Allred and Hall [26] was adopted for all laminates involving Kevlar fibers. In this context, Table 2 presents the results obtained in terms of fibers weight content (wt.%).

## 3. Results

Static bending tests were performed to obtain the hybridization effect on the bending properties. However, to understand the behavior of each fiber in the bending performance, Figure 1 shows typical bending stress/strain curves for non-hybrid configurations (8G, 8K, and 8C), which are representative of all obtained for each configuration analyzed.

It is possible to observe that the 8C and 8G curves practically follow a quasi-linear regime until failure, where the maximum stress occurs, typical behavior of fragile materials. On the other hand, the curve for the Kevlar reinforced composite begins with a quasi-linear regime followed by a nonlinear region in which the maximum stress occurs. This behavior highlights the ductile nature of these fibers. The maximum bending stress is obtained for the carbon/epoxy composite and the smallest for the Kevlar/epoxy composite, with a difference of around 55.2%. The glass/epoxy composite lies between the two, with a bending stress about 25% less than that seen for the carbon/epoxy composite. In terms of bending modulus, the highest value was obtained for the carbon/epoxy composite, with an average value of 48.4 GPa, while the glass/epoxy composite showed a decrease of about 54.3% and the Kevlar/epoxy composite around 56.6%. Regarding the bending strain, the highest value was observed for the Kevlar/epoxy composite, with an average value of 6.2%, while the glass/epoxy and carbon/epoxy composites showed decreases of 46.8% and 67.7%, respectively. The Kevlar/epoxy composite has the lowest bending stress and stiffness but the highest strain, evidencing the ductile nature of the fibers that are classified as high-elongation fibers [27,28]. Finally, the results described above are supported by several studies that evoked similar conclusions [12,13,14,23,24,25,26,27,28] and are a consequence of the different damage mechanisms.

For the different configurations analyzed, Figure 2 shows the main damage mechanisms observed. Regarding the carbon/epoxy composite, Figure 2a evidences fiber fracture in the compression side with quite small delaminations around the broken fibers, which is in good agreement with the open literature [10,29,30,31,32,33]. According to Reis et al. [32], the high compressive stress concentration in the pin load contact region associated with the low compressive strength of the fibers favors the fiber breakage in this region. In terms of the glass/epoxy composite, Figure 2b shows the damage mechanisms observed for this configuration. This laminate, in addition to having higher flexural capacity compared to the carbon laminate, shows that the main damage mechanism occurs with the rupture of the tensile fibers, although some delaminations appear in the pin load contact region due to the stress concentrations that were mentioned earlier (and conveniently reported in [32]). Finally, for the Kevlar/epoxy composite, Figure 2c shows the failure mechanisms, where the large delaminations observed are responsible by the bending strength loss. This failure mode agrees with the study developed by Ferreira et al. [25].

Figure 3 shows the hybridization effect, involving only carbon and glass fibers, on the bending stress/strain curves.

These typical curves represent the behavior of all of them and, similar to the non-hybrid composites (8G and 8C laminates), all curves show an almost linear regime up to the maximum stress where the failure occurs. The damage mechanisms are also similar to those described in Figure 2 and are shown in Figure 4.

It is possible to notice that the specimens involving the carbon/glass composite (4C/4G) present higher damages in the pin load contact region, breakage of the carbon fibers, and some delaminations, while the glass fibers are slightly affected (fractures of punctual fibers and slight delaminations). On the other hand, for the glass/carbon composite (4G/4C), the damage is more severe, involving the breakage of carbon fibers in the tensile region and glass fibers in the compression region. In both cases, the high stress concentration in the pin load contact region significantly affects the local stress field and, consequently, the severity of the damage compared to what would be expected [32]. Consequently, the bending properties change as shown in Table 3, where the mean values and respective standard deviation are presented.

In terms of bending stress, compared to the all-carbon fiber composite (where the average value is around 843.3 MPa), it is possible to note that the increase in the glass fiber content on the compression side promotes a decrease in the bending stress, reaching values about 17.6% lower for the 6G + 2C configuration. However, for the 2G + 6C configuration, the observed decrease was only 2.8%, revealing the benefits reported by Dong et al. [16], where the highest flexural strength was obtained for 12.5% of glass fiber placed on the compressive side. Another evidence that should be highlighted is the fact that when the same configurations have carbon fibers on the compression side, they have much lower values. For example, comparing the 6G + 2C and 6C + 2G configurations, the latter has 6% less bending strength. Finally, all configurations with carbon fibers on the compression side have bending strength values very close to those observed for the all-glass fiber composite (where the average value is around 632.5 MPa). These results confirm the studies developed by Giancaspro et al. [10] and Sudarisman et al. [15], according to which glass fibers perform better on the compression side, while carbon fibers perform better on the traction side. In terms of bending modulus, similar behavior can be observed. While the value obtained for all-carbon fiber composite is around 48.4 GPa, the higher glass fiber content on the compression side promotes a lower bending modulus. Compared to the all-carbon fiber composite, the 6G + 2C configuration has a decrease of 36.7%. On the other hand, for the same configurations, except for 6C + 2G (where the modulus is 17.9% higher than for 6G + 2C), all of the others have lower values when the carbon fibers are placed on the compression side. Concerning the bending strain, the lowest value is observed for all-carbon fiber composite (2%), while the highest value is obtained for the all-glass fiber composite (3.3%). In terms of hybrid configurations, the values obtained are between these, but for the same configurations, those containing carbon fibers on the compression side have smaller strain values.

Regarding the hybridization effect involving only glass and Kevlar fibers, Figure 5 shows typical bending stress/strain curves, which are representative of all others. Similar to the non-hybrid composites (8G and 8K laminates), two behaviors can be found. One is characterized by an almost linear regime up to the maximum stress where the failure occurs, and the other one is characterized by a quasi-linear regime followed by a nonlinear region in which the maximum stress occurs. While the first is typical of the hybrid laminates containing glass fibers on the compression side and the all-glass fiber composite, the second behavior characterizes hybrid laminates containing Kevlar fibers on the compression side and all-Kevlar fibers composite. This is explained by the different damage mechanisms observed in Figure 6. There is noticed, for the 4K/4G configuration, breakage of the glass fibers on the tensile side with some delaminations, while the Kevlar fibers are unaffected (Figure 6a). On the other hand, for the 4G/4K configuration, the breakage of the glass fibers now occurs on the compression side, while the Kevlar fibers remain unchanged (Figure 6b). Once again, in both cases, the high stress concentration in the pin load contact region affects the local stress field, but, due to the intrinsic characteristics of the fibers, the damage is more severe for the 4G/4K configuration. Finally, everything that has been described above can be, as shown in Figure 7 and Figure 8, fully replicated for the hybrid carbon/Kevlar composites but replacing the glass fibers by the carbon fibers.

Table 4 and Table 5 present the bending properties obtained for hybrid composites involving glass/Kevlar fibers and carbon/Kevlar fibers, respectively. Regarding the bending stress, and for the hybrid composite with glass/Kevlar fibers, it is noticed that the increase in the Kevlar fiber content on the compression side promotes a significant decrease, achieving in the 6K + 2G configuration values 47.5% lower than those obtained for the all-glass fiber composite. A similar increase in the bending stress is observed when the glass fiber content increases on the compression site but with higher values when the same configurations are compared. For example, comparing the 6K + 2G and 6G + 2K configurations, the latter has a bending stress 52.3% higher than the 6K + 2G configuration. The same analysis leads to the same conclusions for the hybrid composite involving carbon and Kevlar fibers, where the comparison between the 6K + 2C and 6C + 2K configurations reports, in this case, a difference of only 38.1%, which is 27.2% lower than that observed for the glass/Kevlar composites (between the 6K + 2G and 6G + 2K configurations). At the level of bending modulus, the hybridization effect with glass and Kevlar fibers does not promote significant differences between the different configurations/contents of fibers studied. For example, between the 6K + 2G and 6G + 2K configurations, there is a difference of only 12%. In relation to carbon/Kevlar fibers, and regardless of the position of the fibers, it is possible to verify that a higher level of hybridization promotes higher values of bending modulus. However, when the carbon fibers are placed on the compression side, higher modulus values are also obtained. For example, comparing the 6K + 2C and 6C + 2K configurations, the latter is 46% higher. These results agree with the open literature, where several studies evidence that more layers of glass fibers placed on the compression side promote an increase in strength and modulus [10,11,15,34]. On the other hand, when Kevlar fibers are on the tensile side, due to their strength, they allow the laminate to have less strain and more bending strength [23]. On the other hand, as noted, more Kevlar layers on the compression side and less fiberglass on the tensile side promotes faster laminate fracture, which is also valid for carbon fibers [17].

From the open literature, in general, the hybridization improves the interlaminar shear strength. In this context, for all configurations previously analyzed, the hybridization’s effect on the interlaminar shear strength will be evaluated/analyzed. For this purpose, Table 6 summarizes the ILSS values for full-fiber composites.

It is possible to observe that the highest ILSS value is obtained for composites reinforced with carbon fibers, with an average value of 53.6 MPa, and the lowest for Kevlar/epoxy composites, whose average value is 46.6% lower. Regarding the glass/epoxy composites, the ILSS value is in between these two. The ILSS obtained compared to carbon/epoxy composites is 12.7% lower but 63.6% higher than that observed for Kevlar/epoxy composites. These results are in line with the bibliography, in which it is reported that, regardless of the matrices, Kevlar fibers present weak interfaces, but in relation to the thermoset ones, epoxy resins generally lead to higher ILSS values [27]. According to Madhavi et al. [14], the ILSS values for carbon-reinforced composites are five times higher than those observed for glass fiber-reinforced composites. However, in this study the difference is only 1.15 times higher, but the value obtained is within the range of those available in the literature. For CFRP (carbon fiber-reinforced polymer), GFRP (glass fiber-reinforced polymer), and KFRP (Kevlar fiber-reinforced polymer) composites based on epoxy resins, the open literature reports that the interlaminar shear strength can vary between 40 MPa to 100 MPa [35,36], 40 MPa to 85 MPa [37,38,39], and 28 MPa to 53 MPa [23,38,40,41], respectively, large discrepancies that can be explained by the different properties of the composites’ constituents as well as the different fiber/matrix interfacial adhesions resulting from different fiber surface treatments [42,43,44]. In addition, the different fibers and their orientation have a strong influence on the interlaminar shear strength [14,45].

Regarding the hybridization effect on the interlaminar shear strength, the ILSS values obtained for the different configurations analyzed are summarized in Table 7, Table 8 and Table 9. In fact, according to the literature, the hybridization can improve the interlaminar shear strength. For example, studies developed by Turla et al. [22] showed that the ILSS of hybrid composites is higher than that of glass or carbon composites. However, for all configurations analyzed, this benefit is not observed in the present study. It is noticed that the highest ILSS values are obtained for configurations involving carbon and glass fibers, while the lowest are related to Kevlar fibers. This is explained by the literature, due to the good fiber/matrix adhesion that glass and carbon fibers have [20,22], while Kevlar fibers are recognized for establishing a weak fiber/matrix adhesion [3,46]. Another evidence that can be highlighted is the fact that, in addition to the type of fiber, its quantity and location (compression or tensile side) also affect the interlaminar strength. However, Padmanabhan et al. [23] suggested that the bending modulus is directly related to interlaminar strength and, in this context, an increase in any one of these properties leads to an increase in the other. Simultaneously, Padmanabhan et al. [23] found that ILSS values increase with increasing thickness, due to higher bending stiffness.

Therefore, in this context, due to the hybridization characteristic adopted (layers of two different fibers are stacked onto each other), the fiber content intrinsically affects the thickness of the sublayers and, consequently, the bending modulus of the laminate. On the other hand, it was also noted that the mechanical properties, especially the bending modulus, are strongly affected by the positioning of the fibers in the laminate. Therefore, based on what was previously reported and according to Padmanabhan et al. [23], these parameters justify the different ILSS values obtained and summarized in Table 3, Table 4 and Table 5.

## 4. Conclusions

The main goal of this study was to analyze the bending and interlaminar shear strength of hybrid composites. For this purpose, carbon, glass, and Kevlar fibers were combined with different fiber contents and placed in very specific positions. From this study, it was possible to conclude that:
-for non-hybrid composites, the maximum bending stress and modulus were obtained for the carbon/epoxy composite, while the bending strain was the smallest. On the opposite side, the Kevlar/epoxy composite showed the lowest bending stress and modulus, while the bending strain had the highest value. These results were explained by the intrinsic properties of the composites’ constituents and by the damage mechanisms that proved to be very specific for each laminate. The interlaminar shear strength followed the same trend, with the highest ILSS value for the carbon/epoxy composite and the lowest for the Kevlar/epoxy composite;-about the hybridization effect, the highest values were obtained for composites involving carbon and glass fibers, with the latter placed on the compression side. This proves the poor compression performance of the carbon fibers. On the other hand, the results were very similar when Kevlar fibers were placed on the tensile side. Finally, the highest ILSS values were obtained for the composite involving carbon and glass fibers, while the lowest ILSS values were obtained for composites involving Kevlar fibers. Furthermore, it was observed that the fiber content and its positioning in the laminate affect both flexural strength and interlaminar shear strength, evidencing that these properties may be related.

## Figures and Tables

**Figure 1 materials-15-01302-f001:**
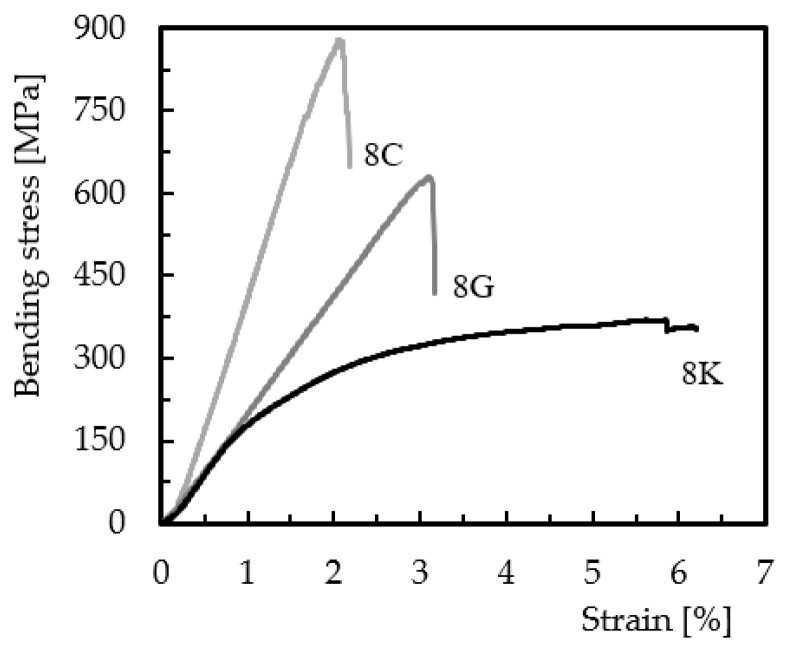
Bending stress/strain curves for full fiber composites.

**Figure 2 materials-15-01302-f002:**
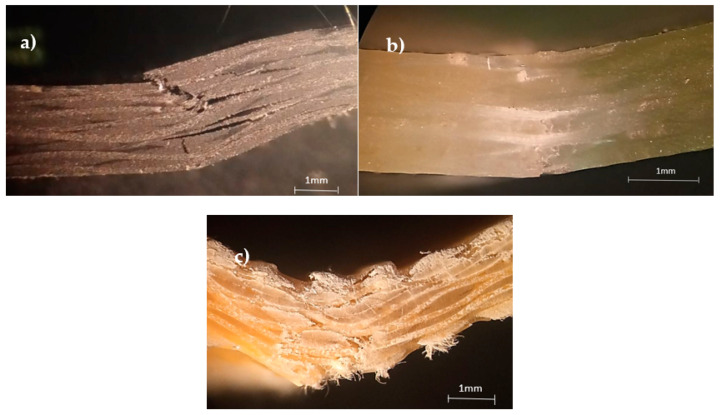
Damage mechanisms observed for: (**a**) carbon/epoxy composites; (**b**) glass/epoxy composites; (**c**) Kevlar/epoxy composites.

**Figure 3 materials-15-01302-f003:**
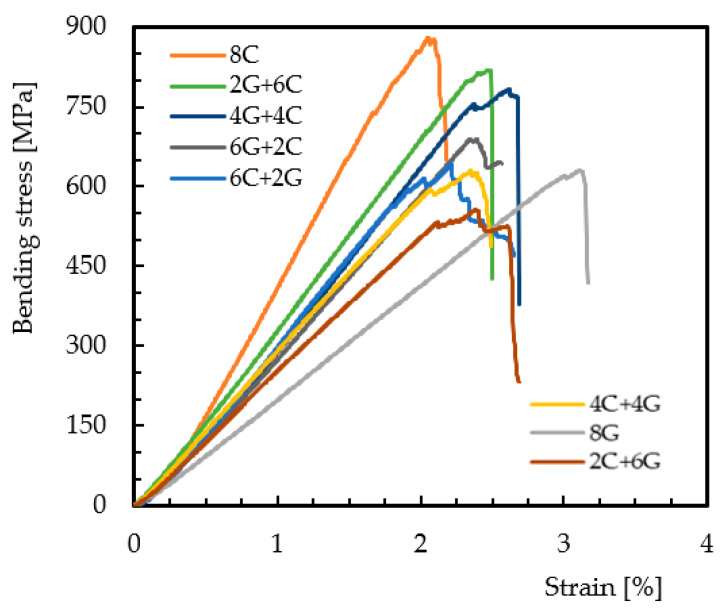
Bending stress/strain curves for hybrid composites involving carbon and glass fibers.

**Figure 4 materials-15-01302-f004:**
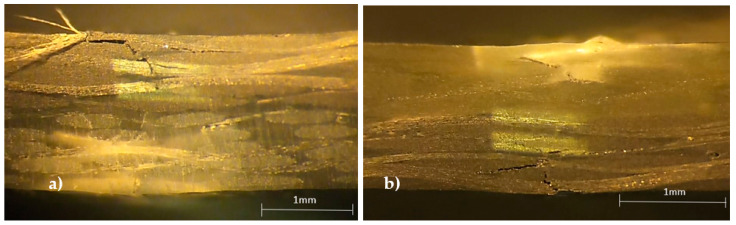
Damage mechanisms observed for: (**a**) carbon/glass specimens (4C/4G); (**b**) glass/carbon specimens (4G/4C).

**Figure 5 materials-15-01302-f005:**
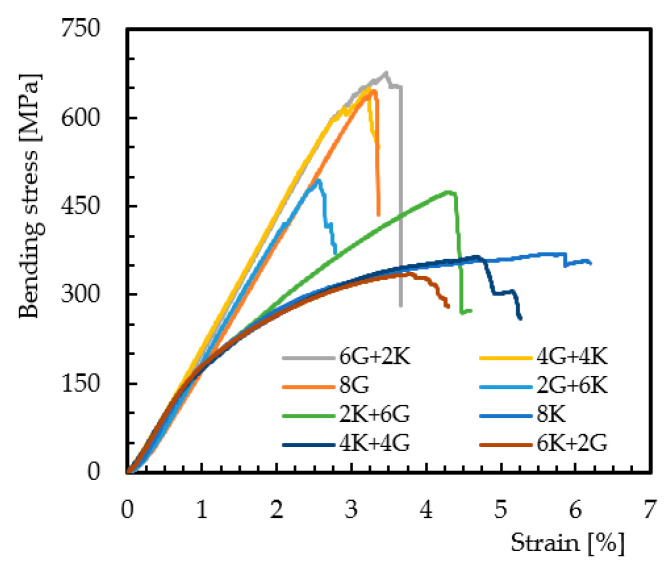
Bending stress/strain curves for hybrid composites involving glass and Kevlar fibers.

**Figure 6 materials-15-01302-f006:**
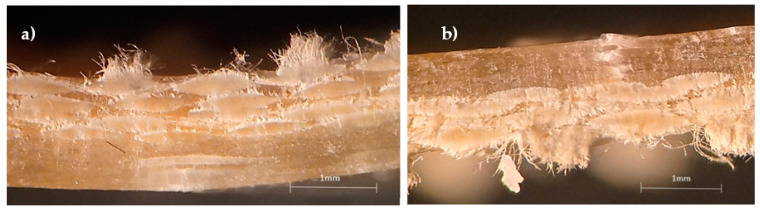
Damage mechanisms observed for: (**a**) Kevlar/glass specimens (4K/4G); (**b**) glass/Kevlar specimens (4G/4K).

**Figure 7 materials-15-01302-f007:**
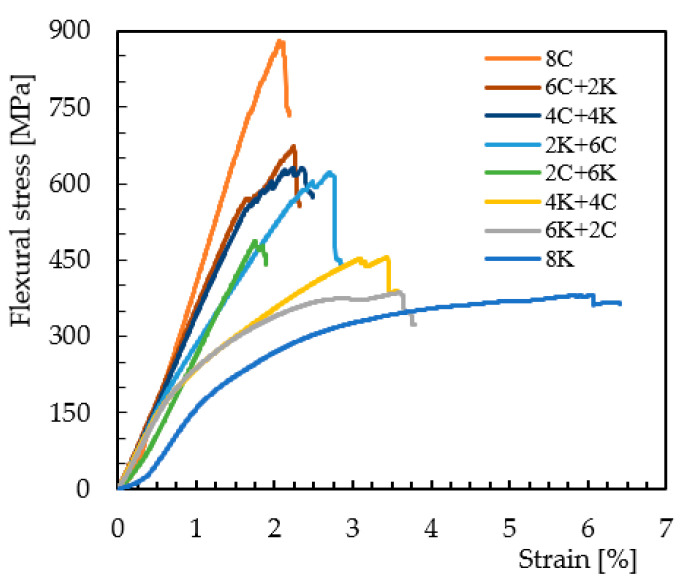
Bending stress/strain curves for hybrid composites involving carbon and Kevlar fibers.

**Figure 8 materials-15-01302-f008:**
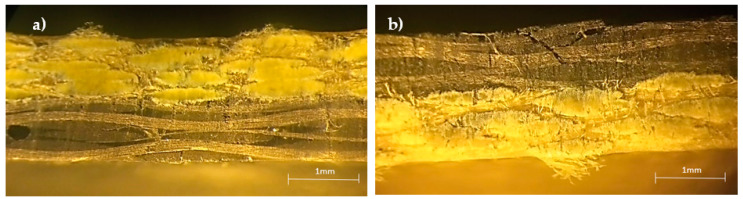
Damage mechanisms observed for: (**a**) Kevlar/carbon specimens (4K/4C); (**b**) carbon/Kevlar specimens (4C/4K).

**Table 1 materials-15-01302-t001:** Sample stacking sequence and respective average thickness.

Group 1	Average Thickness [mm]	Group 2	Average Thickness [mm]	Group 3	Average Thickness [mm]
8C	1.8	8G	1.5	8K	1.9
2C + 6G	1.6	2G + 6K	1.8	2K + 6C	1.8
4G + 4C	1.7	4G + 4K	1.7	4K + 4C	1.9
6C + 2G	1.7	6G + 2K	1.6	6K + 2C	1.9

**Table 2 materials-15-01302-t002:** Fiber content (wt.%) for the different composite laminates.

Laminates	Fiber Content (wt.%)		
	Carbon Fibers	Glass Fibers	Kevlar Fibers
8C	60 ± 0.23	-	-
2C + 6G	15 ± 0.19	49 ± 0.18	-
4C + 4G	30 ± 0.16	32 ± 0.79	-
6C + 2G	43 ± 0.26	17 ± 0.28	-
8G	-	63 ± 0.19	-
2G + 6K	-	17 ± 0.97	41 ± 0.22
4G + 4K	-	32 ± 0.48	28 ± 0.36
6G + 2K	-	49 ± 0.25	13 ± 0.14
8K	-	-	56 ± 0.2
2K + 6C	43 ± 0.58	-	13 ± 0.17
4K + 4C	30 ± 0.25	-	28 ± 0.25
6K + 2C	15 ± 0.22	-	41 ± 0.19

**Table 3 materials-15-01302-t003:** Bending properties for hybrid composites involving carbon and glass fibers.

Laminates	σ_f_ [MPa]	E_f_ [GPa]	E
8C	843.3 ± 33.2	48.4 ± 0.9	2.0 ± 0.06
2G + 6C	820.0 ± 35.0	38.8 ± 0.6	2.5 ± 0.10
4G + 4C	785.2 ± 20.5	34.8 ± 1.9	2.6 ± 0.08
6G + 2C	694.6 ± 16.8	30.7 ± 1.6	2.4 ± 0.07
6C + 2G	652.9 ± 27.6	37.4 ± 4.4	2.2 ± 0.10
4C + 4G	637.2 ± 12.8	31.0 ± 2.8	2.4 ± 0.09
2C + 6G	596.1 ± 25.1	27.7 ± 2.8	2.5 ± 0.08
8G	632.5 ± 11.8	22.1 ± 0.7	3.3 ± 0.09

**Table 4 materials-15-01302-t004:** Bending properties for hybrid composites involving glass and Kevlar fibers.

Laminates	σ_f_ [MPa]	E_f_ [GPa]	ε_f_ [%]
8G	632.5 ± 11.8	22.1 ± 0.7	3.3 ± 0.09
2K + 6G	465.4 ± 16.0	19.0 ± 0.9	4.4 ± 0.08
4K + 4G	354.1 ± 13.1	19.9 ± 1.5	4.6 ± 0.30
6K + 2G	332.3 ± 7.0	20.5 ± 2.2	4.0 ± 0.13
6G + 2K	696.5 ± 41.2	23.3 ± 1.5	3.4 ± 0.10
4G + 4K	646.8 ± 20.3	23.2 ± 1.4	3.2 ± 0.16
2G + 6K	500.6 ± 14.0	23.9 ± 1.8	2.4 ± 0.22
8K	378.2 ± 8.8	21.0 ± 1.3	6.2 ± 0.46

**Table 5 materials-15-01302-t005:** Bending properties for hybrid composites involving carbon and Kevlar fibers.

Laminates	σ_f_ [MPa]	E_f_ [GPa]	ε_f_ [%]
8C	843.3 ± 33.2	48.4 ± 0.9	2.0 ± 0.06
2K + 6C	629.4 ± 32.2	24.2 ± 1.8	2.6 ± 0.19
4K + 4C	471.4 ± 18.3	25.6 ± 1.8	3.4 ± 0.18
6K + 2C	399.0 ± 20.1	29.1 ± 3.5	3.8 ± 0.30
6C + 2K	644.6 ± 32.0	42.5 ± 6.0	2.0 ± 0.20
4C + 4K	638.7 ± 19.0	34.5 ± 2.4	2.0 ± 0.09
2C + 6K	488.8 ± 34.7	29.9 ± 2.9	1.8 ± 0.06
8K	378.2 ± 8.8	21.0 ± 1.3	6.2 ± 0.46

**Table 6 materials-15-01302-t006:** ILSS for full-fiber composites.

Laminates	ILSS [MPa]	Decrease in Relation to 8C [%]
8C	53.6 ± 1.3	-
8G	46.8 ± 2.2	−12.7%
8K	28.6 ± 1.1	−46.6%

**Table 7 materials-15-01302-t007:** ILSS for hybrid composites involving carbon and glass fibers.

Laminates	ILSS [MPa]	Decrease in Relation to 8C [%]
8C	53.6 ± 1.29	-
2G + 6C	46.7 ± 0.96	−12.9%
4G + 4C	51.1 ± 0.58	−4.7%
6G + 2C	45.3 ± 2.10	−15.5%
6C + 2G	45.6 ± 1.04	−14.9%
4C + 4G	48.9 ± 1.98	−8.8%
2C + 6G	39.6 ± 2.91	−26.1%
8G	46.8 ± 2.15	−12.7%

**Table 8 materials-15-01302-t008:** ILSS for hybrid composites involving glass and Kevlar fibers.

Laminates	ILSS [MPa]	Decrease in Relation to 8G [%]
8G	46.8 ± 2.15	-
2K + 6G	34.5 ± 4.05	−26.3%
4K + 4G	30.8 ± 2.22	−34.2%
6K + 2G	33.2 ± 3.21	−29.1%
6G + 2K	37.6 ± 1.15	−19.7%
4G + 4K	34.4 ± 0.81	−26.5%
2G + 6K	32.5 ± 0.98	−30.6%
8K	28.6 ± 1.09	−38.9%

**Table 9 materials-15-01302-t009:** ILSS for hybrid composites involving carbon and Kevlar fibers.

Laminates	ILSS [MPa]	Decrease in Relation to 8C [%]
8C	53.6 ± 1.29	-
2K + 6C	44.8 ± 0.58	−16.4%
4K + 4C	34.5 ± 1.57	−35.6%
6K + 2C	31.9 ± 0.7	−40.5%
6C + 2K	42.5 ± 1.53	−20.7%
4C + 4K	37.0 ± 1.69	−31.0%
2C + 6K	36.2 ± 0.82	−32.5%
8K	28.6 ± 1.09	−46.6%

## Data Availability

Not applicable.

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
