# Peer review of "Hybridization Effects on Bending and Interlaminar Shear Strength of Composite Laminates"

_materials, 2022, doi:10.3390/ma15041302_

Round 1

Reviewer 1 Report

This paper reports on the “Hybridization effects on bending and interlaminar shear strength of composite laminates”. Introduction and conclusion, methodology and reference, results and discussion seems be corrected.

I have few comments to the manuscript:

  1. All manuscript. Corrected from e.g. “[1,2]” to “[1-2]”.
  2. Goal of work should be clear presented.
  3. Separated In chapter 2 Materials, used methods and description of obtained samples. Description of methods is too short.
  4. Add standard deviations.

Taking into account all comments the manuscript may be published in Materials after minor revision.

Author Response

Response to Reviewer Nº. 1

This paper reports on the “Hybridization effects on bending and interlaminar shear strength of composite laminates”. Introduction and conclusion, methodology and reference, results and discussion seems be corrected.

Response: Thank you for your comments and appreciation of our work.

1 - All manuscript. Corrected from e.g. “[1,2]” to “[1-2]”.

Response: The text was improved according to the reviewer’s suggestion.

2 - Goal of work should be clear presented.

Response: According to the reviewer’s suggestion, the text was improved for “… Nevertheless, according to Swolfs et al. [5], hybrid effects under more complex loading conditions, such as in bending, impact and fatigue tests, are not well understood and sometimes even promote apparent contradictions. These authors even suggest further work to obtain more robust conclusions. Therefore, this work intends to consolidate the conclusions presented in the literature in terms of static bending strength and interlaminar shear strength. For this purpose, composites involving different fibres (such as carbon, glass and Kevlar fibres) and different values of weight content were used …”.

3 -   Separated In chapter 2 Materials, used methods and description of obtained samples. Description of methods is too short.

Response: Authors improved section 2 as suggested by the reviewer. Authors do not separate this section into different subsections because some of them would be so small that they are not large enough for a subsection. Finally, methods were improved (see the improvements in the text with blue colour).

4 - Add standard deviations.

Response: As suggested by the reviewer, authors included the standard deviations for all experimental results presented in a table format.

Reviewer 2 Report

The authors addressed one of the main weak points of fiber-reinforced composite materials, which is the lack of toughness by employing a new strategy, hybridization. They used a different binary combination of carbon, glass, and Kevlar fibers to improve the properties of an AH 150 resin. They studied the hybridization effect on the static performance and interlaminar shear strength. Although some improvements were observed in the K/G composite samples, the improvements were not significant in the C-based samples. It seems all the properties except bending strain were the best in the plain caron fiber reinforced samples. Accordingly, the authors need to explain the main benefit of hybridization. Some minor comments can be found in the following.

C1. It is recommended to include some of the obtained results in the Abstract.

C2. It would be interesting for the readers to know the concentration of the fibers in the resin matrix. Furthermore, it is necessary to report the orientation of the fibers in the epoxy matrix.

C3. It is recommended to report the std Dev as ± attached to the reported average value. Furthermore, Table 2 can be removed since the data of the full fiber composites have been reported in other tables.

C4. The technique used for fiber imaging needs to be introduced in the methods section.

C5. Almost no citations can be seen from recent years.

Author Response

Response to Reviewer Nº. 2

The authors addressed one of the main weak points of fiber-reinforced composite materials, which is the lack of toughness by employing a new strategy, hybridization. They used a different binary combination of carbon, glass, and Kevlar fibers to improve the properties of an AH 150 resin. They studied the hybridization effect on the static performance and interlaminar shear strength. Although some improvements were observed in the K/G composite samples, the improvements were not significant in the C-based samples. It seems all the properties except bending strain were the best in the plain caron fiber reinforced samples. Accordingly, the authors need to explain the main benefit of hybridization. Some minor comments can be found in the following.

Response: Thank you for your comments and appreciation of our work.

1 - It is recommended to include some of the obtained results in the Abstract.

Response: As suggested by the reviewer, authors improved the Abstract by including some experimental results.

2 -   It would be interesting for the readers to know the concentration of the fibers in the resin matrix. Furthermore, it is necessary to report the orientation of the fibers in the epoxy matrix.

Response: According to the reviewer’s suggestion, authors included the fiber content for the different composites, as evidenced by the new Table 2 inserted in the text, as well as the fiber orientation (“… kevlar fibre woven bidirectional fabric (taffeta with 170 g/m2), all in the same direction, with an Ebalta AH 150 resin and IP 430 hardener were used to prepare different composite laminates …”).

3 -   It is recommended to report the std Dev as ± attached to the reported average value. Furthermore, Table 2 can be removed since the data of the full fiber composites have been reported in other tables.

Response: As suggested by the reviewer, the standard deviation (in ± format) was included in the Tables, as well as Table 2 was amended with other data.

4 - The technique used for fiber imaging needs to be introduced in the methods section.

Response: As suggested by the reviewer, authors included a sentence to describe the technique used in the analysis of failure modes (The failure modes resulting from the bending tests performed for the different configurations were analysed using a Nikon optical microscope (model SMZ-2T).).

5 - Almost no citations can be seen from recent years.

Response: As suggested by the reviewer, authors included references from recent years.

Reviewer 3 Report

1. In equation (1) it should be σ= 3PL/2bh2 instead of σ= 3PL/ 2bh.

2. In equation (2) it should be E = PL/ 48∆uI instead of E = ∆PL /48∆uI.

3. It is necessary to use symbols from Eq.1 and Eq.2 in the Table 2 for bending stress and for bending stiffness. How was the bend deformation (bending strain) determined? It is desirable to add a suitable equation.

4. Tables 3,4 and 5 may be demonstrated in a more compact form. It is possible to use only one column for symbols of a laminates composition (like8C, 2K+6C, 4K+4C etc. ).

5. Conclusions should be shorter (about 4-8 times).

Author Response

Response to Reviewer Nº. 3

 1 - In equation (1) it should be σ= 3PL/2bh2 instead of σ= 3PL/ 2bh.

Response: The text was improved according to the reviewer’s suggestion. Authors apologize for the mistake contained in the document.

2 -   In equation (2) it should be E = PL/ 48∆uI instead of E = ∆PL /48∆uI.

Response: The equation was validated by the ASTM D790-03 standard and the following sentence was included. “… The bending modulus was obtained by linear regression of the load-displacement curves considering the interval in the linear segment with a correlation factor higher than 95% …”.

3 -   It is necessary to use symbols from Eq.1 and Eq.2 in the Table 2 for bending stress and for bending stiffness. How was the bend deformation (bending strain) determined? It is desirable to add a suitable equation.

Response: As suggested by the reviewer, the symbols were included in the Tables by the authors. In addition, the bending strain equation was also included in the document.

4 -   Tables 3,4 and 5 may be demonstrated in a more compact form. It is possible to use only one column for symbols of a laminates composition (like 8C, 2K+6C, 4K+4C etc.).

Response: As suggested by the reviewer, authors presented a more compact version of the Tables.

5 - Conclusions should be shorter (about 4-8 times).

Response: As suggested by the reviewer, conclusions have been revised in order to reduce the text.